# Glucose/Ribitol Dehydrogenase and 16.9 kDa Class I Heat Shock Protein 1 as Novel Wheat Allergens in Baker’s Respiratory Allergy

**DOI:** 10.3390/molecules27041212

**Published:** 2022-02-11

**Authors:** Mario Olivieri, Gianluca Spiteri, Jessica Brandi, Daniela Cecconi, Marina Fusi, Giovanna Zanoni, Corrado Rizzi

**Affiliations:** 1Unit of Occupational Medicine, Department of Diagnostic and Public Health, University of Verona, 37134 Verona, Italy; mario.olivieri@univr.it (M.O.); gianluca.spiteri@univr.it (G.S.); 2Department of Biotechnology, University of Verona, 37134 Verona, Italy; jessica.brandi@univr.it (J.B.); daniela.cecconi@univr.it (D.C.); marina.fusi@gmail.com (M.F.); 3Immunology Unit, University Hospital of Verona, 37135 Verona, Italy; giovanna.zanoni@univr.it

**Keywords:** GRDH, HSP 16.9, baker’s asthma and rhinitis, mass spectrometry, proteomics

## Abstract

Wheat allergens are responsible for symptoms in 60–70% of bakers with work-related allergy, and knowledge, at the molecular level, of this disorder is progressively accumulating. The aim of the present study is to investigate the panel of wheat IgE positivity in allergic Italian bakers, evaluating a possible contribution of novel wheat allergens included in the water/salt soluble fraction. The water/salt-soluble wheat flour proteins from the Italian wheat cultivar Bolero were separated by using 1-DE and 2-DE gel electrophoresis. IgE-binding proteins were detected using the pooled sera of 26 wheat allergic bakers by immunoblotting and directly recognized in Coomassie stained gel. After a preparative electrophoretic step, two enriched fractions were furtherly separated in 2-DE allowing for detection, by Coomassie, of three different proteins in the range of 21–27 kDa that were recognized by the pooled baker’s IgE. Recovered spots were analyzed by nanoHPLC Chip tandem mass spectrometry (MS/MS). The immunodetected spots in 2D were subjected to mass spectrometry (MS) analysis identifying two new allergenic proteins: a glucose/ribitol dehydrogenase and a 16.9 kDa class I heat shock protein 1. Mass spectrometer testing of flour proteins of the wheat cultivars utilized by allergic bakers improves the identification of until now unknown occupational wheat allergens.

## 1. Introduction

Baker’s asthma and rhinitis are frequent occupational diseases with an estimated annual incidence ranging from 1 to 10 cases per 1000 bakery workers and its prevalence does not seem to be declining [1,2,3]. Any workers exposed to airborne cereal flours inhalation could develop these occupational allergic diseases so 60–70% of bakers with work-related allergic symptoms have increased specific IgE levels to cereal flours [4,5].

The causing agents are proteins mainly present in wheat flour; although, additives and enzymes contained in multigrains might be involved [6].

Wheat proteins can be classified according to their solubility in three fractions: (i) the water/salt-soluble fraction (WSSF) including albumins and globulins, (ii) the water/ethanol-soluble gliadins, and (iii) the glutenins, with the latter soluble in weak acids [7].

Actually, in wheat proteins resolved by 2D electrophoresis, more than 100 IgE-binding spots have been detected utilizing the sera of sensitized workers, suggesting high variability in individual sensitization to wheat proteins [8].

The highest percentage of IgE binding was observed in the WSSF containing several allergens identified by proteomic approach even if all the three wheat flour fractions contain IgE reactive proteins [8,9,10]. This complicates the clinical definition of the allergic bakers’ profile affected by work-related diseases where diagnosis and therapy are crucial steps. A lot of studies have been progressively focused on the identification and characterization of new wheat allergens involved in bakers’ allergies; although, not a single allergen emerged as major one due to the interindividual IgE-binding profile as shown in Table 1 [7,11,12,13,14,15,16,17,18,19,20,21,22].

Wheat allergic sensitization evaluated by skin prick test (SPT) or by IgE positivity to wheat extracts, has a higher diagnostic sensitivity when compared with component-resolved diagnosis of baker’s sensitization [23]. Besides, SPT does not evaluate the specific role of each wheat allergens in different occupational diseases. The aim of the present study was to investigate the panel of wheat IgE positivity in allergic Italian bakers in order to evaluate a possible contribution of unidentified wheat allergens. For this purpose, we analyzed wheat water-salt soluble proteins from the cultivar Bolero, a wheat widely used in breadmaking in Italy [24].
molecules-27-01212-t001_Table 1Table 1Biochemical activities and route of exposure of wheat allergens.AllergenBiochemical ActivityExposureRef.Ethylene-responsive transcription factor Plant transcription factorInhalation[12]Transcription elongation factor 1 Plant transcription factorInhalation[12]Transcription factor with zinc finger domainPlant transcription factorInhalation[12]Translation factor 5a2 Translation factorInhalation[12]Tri a 15Monomeric alpha-amylase inhibitor 0.28Inhalation[11]Tri a 29Tetrameric alpha-amylase inhibitor CM1/CM2Inhalation[11]Tri a 30Tetrameric alpha-amylase inhibitor CM3Inhalation[11]Tri a 31Triosephosphate-isomeraseInhalation[7]Tri a 321-cys-peroxiredoxinInhalation[16]Tri a 33SerpinInhalation[7]Tri a 34Glyceraldehyde-3-phosphate-dehydrogenaseInhalation[20]Tri a 35DehydrinInhalation[20]Tri a 39Serine protease inhibitor-like proteinInhalation[16]Tri a 40Chloroform/methanol-soluble (CM) 17 protein [alpha-amylase inhibitor]Inhalation[11]26 kDa endochitinase EndochitinaseIngestion[21]b-glucosidase GlucosidaseIngestion[21]Class II chitinase ChitinaseIngestion[21]Endogenous amylase/subtilisin inhibitor Alpha amylase/subtilisin inhibitorIngestion[21]TLP Thaumatin like proteinIngestion[21]Tri a 12ProfilinIngestion[17]Tri a 14Non-specific lipid transfer protein 1Ingestion[14]Tri a 17beta-amylaseIngestion[17]Tri a 18Agglutinin isolectin 1Ingestion[14]Tri a 19Omega-5 gliadin, seed storage proteinIngestion[22]Tri a 20Gamma gliadinIngestion[15]Tri a 21Alpha-beta-gliadinIngestion[15]Tri a 25ThioredoxinIngestion[14]Tri a 26High molecular weight gluteninIngestion[14]Tri a 27Thiol reductase homologueIngestion[7]Tri a 28Dimeric alpha-amylase inhibitor 0.19Ingestion[11]Tri a 36Prolamin. Low molecular weight glutenin GluB3-23Ingestion[19]Tri a 36Low molecular weight glutenin GluB3-23Ingestion[13]Tri a 37 a-purothionin. PR-13 familyIngestion[18]Tri a 41Mitochondrial ubiquitin ligase activator of NFKB 1Ingestion[25]Tri a 42Hypothetical protein from cDNAIngestion[25]Tri a 43Hypothetical protein from cDNAIngestion[25]Tri a 44Endosperm transfer cell specific PR60 precursorIngestion[25]Tri a 45Elongation factor 1 (EIF1)Ingestion[25]Trypsin/a-amylase inhibitor (AAI) CMX1/CMX3Tripsin/a-amylase inhibitorIngestion[21]Xylanase inhibitor protein-1 Xylanase inhibitorIngestion[21]


## 2. Results

The clinical characteristics of the twenty-six bakers, 22 males and four females, referring work-related rhinitis only, occupational asthma-like symptoms only and both work-related rhinitis and asthma-like symptoms when exposed to wheat flours in bakeries are reported in Table 2. Only five of them were smokers and obesity (BMI > 30) were detected in seven of them. A positive skin prick test (SPT) (wheals ≥ 3 mm) to any environmental allergen, reported in the Table 2 as atopy, was detected in 20 bakers while a positive SPT or IgE positivity (>0.35 kU/L) to wheat flour was demonstrated in all of them.

### 2.1. The IgE-Binding Patterns of Patients

The sera were at first individually utilized to probe the WSSF separated by SDS-PAGE (Figure 1). The probed lanes are subdivided into three groups according to bakers work-related symptoms: rhinitis only (panel A, lanes 1–11), asthma only (panel B, lanes 12–17), and asthma plus rhinitis (panel C, lanes 18–26). A group of proteins, migrating between molecular masses about from 21 to 35 kDa were mainly detected. Other signals were observed in the 12–16 kDa region. A noticeable inter-individual variability is evident even though some bands seem to be recognized by multiple sera with different intensities, mainly in those with both nasal and asthmatic symptoms in bakeries. As expected, the lanes probed with control sera (C1–C2) were almost completely negative.

### 2.2. 2-DE Identification of Different IgE-Binding Spot in Raw and Fractionated Wheat Extract

We pooled the sera of bakers with work-related wheat respiratory symptoms in order to identify the IgE-binding proteins responsible of the immunoreactivity of the bands detected in Figure 1. WSSF were resolved by 2-DE (panel A Figure 2) and the pooled sera were used to probe the blotted protein spots (panel B Figure 2). After 2D separation (Figure 2, panels A and B) the main recognized protein bands split into several spots recognized by IgE in the 6–7.5 pI range. Among these spots, the most intense (indicated by the arrows) were excised from Coomassie stained gel (Figure 2, panel A), and identified by nanoHPLC Chip S/MS respectively as: Alpha-amylase inhibitor 0.28 (*Triticum aestivum*) with a 12 MW;Glucose/ribitol dehydrogenase (*Oryza sativa*) with the 31 MW.

Alpha-amylase inhibitor 0.28 is a well-known wheat allergen while glucose/ribitol dehydrogenase has never been described in literature as a wheat allergen [20,26]. In addition, no immunoreactive spot is detectable under 10 kDa, while a group of not well resolved spots at 45–60 kDa are strongly immunodetected although the scarce resolution of the spots does not allow a consistent MS identification.

In the same panel, a 21–27 kDa range of migrating proteins split into a train of spots that were not detectable in Coomassie staining, so a preparative electrophoretic separation step was done. After SDS removal from the eluate fractions (arrows in panel A of Figure 3), the WSSF proteins were subjected to a further 2D separation. In this case, the immunodetected spots were stained by Coomassie (data not shown), pricked (arrows in panels B and C of Figure 3), and processed by MS analyses.

The results obtained by nanoHPLC Chip MS/MS analysis of the spots 1–2 (Figure 2) and 3–5 (Figure 3) are summarized in Table A1 (Appendix A). Besides the monomeric alpha amylase inhibitor 0.28, two further proteins, triosephosphate-isomerase, a rare cause of baker’s asthma, and thioredoxin-peroxidase, recognized as cause of wheat oral allergy were also identified [8,10,14,27]. Finally, two proteins that were never reported as wheat allergens, the glucose/ribitol dehydrogenase and a 16.9 kDa class I heat shock protein 1, were also identified.

## 3. Discussion

Baker’s asthma and rhinitis are the most frequent form of occupational respiratory diseases, due to a lot of wheat allergens. As well reviewed by Brant, the traditional handcraft bakery is rapidly transforming and, in many countries, new workforces, as supermarkets employees, are identified at risk of inhalant wheat flour allergy [28]. The changes of job task exposure are further associated with technological innovation in bread making as well as the use of multi-grain flours from other cereals or enzymes. These innovations induce a progressive challenge of allergen exposures in bakeries.

When dealing with bakers allergic to wheat flour, other variables should be considered like the specific wheat cultivars utilized in milling, and other factors that could produce different allergen content and expression in flours. Although it seems that there are no clinically significant differences in the allergenicity of wheat of different species, without considering the more than 25,000 cultivars, the different clinical symptoms and allergen-associated could vary with the different wheat species or patients in different regions [29,30]. In addition, different populations may show different responses to allergens in relationship to other environmental factors or genetic characteristics [31].

The knowledge of the specific allergens involved in exposure-related allergic reaction is a prerequisite for the development of diagnostic and therapeutic tools, so we have recruited Italian bakers reporting allergic symptoms and using Italian common wheat flours like the Bolero one.

It was assumed that the most frequently (>50%) IgE-detected-bands could represent the so-called major allergens for the studied population although there is no consensus definition of major and minor allergens of wheat, excluding omega-5 gliadin, that is the wheat protein cause of WDEIA [32]. Accordingly to the literature, our results confirm no IgE-binding specific pattern for work-related allergic respiratory diseases (rhinitis only, asthma only, and rhinitis and asthma) [33].

We have analyzed the proteins contained in water-salt soluble fraction, that is reported to have the strongest IgE reactivity in wheat allergic bakers. These proteins were probed with the sera of 26 bakers with respiratory symptoms due to wheat flour occupational exposure. To identify the major allergens involved we have tested them separately (Figure 1). Among work-related allergic respiratory diseases, more and higher intensity bands signals seem to be detected by sera of bakers referring asthma and rhinitis although the limited number of subjects does not allow further considerations concerning their role in these allergic diseases.

We have then carried out 2-D separations of WSSF and probed by the pooled sera of symptomatic bakers (rhinitis and/or asthma), identifying alpha-amylase inhibitor and glucose/ribitol dehydrogenase (GRDH). Due to the Coomassie staining limit in the identification of proteins that are lowly expressed in wheat but recognized by bakers IgE (Figure 2A,B), we used a preparative electrophoretic step. This procedure, coupled with further 2D maps, allows one to identify two novel wheat allergens, identified as glucose/ribitol dehydrogenase and 16.9 Heat shock protein. The physiological role of these proteins in wheat deserves some considerations. During high-temperature exposure or other stressing conditions plants synthesize many stress-responsive protein families including the so-called heat shock proteins (HSPs) having a role in the refolding and solubilization of denatured protein aggregates [34]. Even the glucose/ribitol dehydrogenase, an enzyme related to glucose degradation, plays a role in desiccation and salinity tolerance in cereal seeds [35].

Thioredoxin peroxidase (TPx) has an antioxidant function removing reactive oxygen species using thioredoxin as the electron donor and an increased TPx activity and expression under drought treatment, thus inducing a delay of drought-induced damage [36,37].

The global warming and rising earth temperatures suggest the importance to deep the knowledge about these proteins expression that provide thermo-tolerance in crop. Indeed, it cannot be excluded that clime changes occurred in last decades may have increased the expression of these defense proteins, thus enhancing the allergenicity of wheat flours [38].

### Strengths and Limitations

The use of a standardized diagnostic protocol in bakers of a well-defined geographical area including mass spectrometry to test wheat proteins allergenicity of a cultivar utilized by allergic bakers in their workplaces.

Our results are not applicable at the individual level of each investigated bakers nor in relation to their pathology due to the analyses of pooled sera so comparison of these results at an individual level are needed particularly in follow-up studies. In addition, our study is biased by the inclusion of symptomatic wheat IgE and/or SPT positive bakers only. Indeed, the proteomic analysis was limited to the WSSF wheat proteins.

## 4. Materials and Methods

### 4.1. Study Patients

The baseline study population was identified through a cross-sectional postal survey of bakery workers in the Verona area in order to evaluate the prevalence of occupational allergic disorders and the relative risk factors [39]. Twenty-six bakers referring exposure to flours at work, occupational respiratory symptoms in relation to wheat exposure and elevated specific IgE antibodies to wheat flour (>0.35 kU/L) or positive skin prick test (SPT) wheals (≥3 mm) to wheat flour were recruited. No one referred allergic symptoms after wheat derivatives ingestion nor history of wheat-dependent exercise-induced anaphylaxis (WDEIA). Two control subjects unexposed and unsensitized to cereal flours were also tested. The study was approved by the Ethics Committee of Verona University Hospital and participants gave written informed consent.

### 4.2. Skin Prick Tests (SPT)

SPT extracts of environmental allergens included pollen (birch, olive tree, grass, Parietaria, Artemisia, and Ambrosia), cat dander, *Dermatophagoides pteronyssinus*, *Dermatophagoides farinae*, *Alternaria alternata*, *Cladosporium herbarum and Blatella germanica,* and wheat flour (Lofarma, Milano, Italy) were tested. Histamine phosphate (10 mg/mL) and normal saline were used as positive and negative controls, respectively. Positive SPT was defined as a wheal diameter ≥3 mm [40]. Atopy was defined by at least one SPT positivity to any environmental allergen.

### 4.3. IgE Determination

Total IgE and specific IgE to wheat flour were performed by ImmunoCAP (Phadia, Uppsala, Sweden). Specific IgE values of 0.35 kU/L or more were considered positive.

### 4.4. Pulmonary Function Testing

Each patient underwent baseline spirometry (SensorMedics V-Max 22, Milan, Italy) according to the ATS/ERS guidelines. In subjects without airway obstruction, according to the definition proposed by ERS for lower limit of normal for FEV1/FVC (88% predicted for men and 89% predicted for women) 30 (Miller 2005), the methacholine challenge (MB3 dosimeter, Mefar, Brescia, Italy) was performed according to guidelines [41]. The methacholine challenge was stopped when a cumulative dose of 2 mg of methacholine was reached or when the forced expiratory value at 1 s (FEV1) had fallen by 20% or more below the best baseline FEV1 following diluent inhalation (PD20 FEV1). The test was considered as positive for a provocative dose of methacholine (PD20 FEV1) ≤ 1 mg [13,41].

### 4.5. Wheat Protein Extraction and Purification

Wheat flour (Bolero cultivar) was provided by Istituto Sperimentale per la Cerealicoltura, Sezione di Sant’Angelo Lodigiano (Lodi, Italy). The water/salt-soluble protein fraction was extracted with phosphate-buffered-saline (PBS: 10 mM sodium dihydrogen phosphate/disodium hydrogen phosphate, 150 mM NaCl, pH 7.2), at 1:10 *w*/*v*, for 2 h under agitation. The samples were centrifuged by a Beckman Coulter Allegra 21R centrifuge (Milano, Italy) (12,000× *g* for 15 min at 4 °C) and the supernatants collected and stored at −20 °C. Protein concentration was determined by bicinchoninic acid assay (BCATM protein assay kit, Pierce, Cheshire, UK) using bovine serum albumin (BSA) as standard [42].

### 4.6. SDS-PAGE and IgE-Immunoblotting

WSSF proteins (50 µg/well) in 0.2 M Tris-HCl, pH 6.8, 10% glycerol, 2% SDS, 5% 2-mercaptoethanol were separated by 16% SDS-PAGE using Mini-Protean III (Bio-Rad, Milano, Italy) at constant 20 mA current. Gels were stained by Coomassie R-250 Blue (Bio-Rad, Milano, Italy) or electroblotted onto 0.45-µm PVDF membrane (Millipore, Milano, Italy) at 50 V × 150 min. After blocking (3% defatted dry milk, 0.05% Tween 20 in PBS) membranes were probed with patients’ sera at 1:10 dilution. Human IgE were detected by a monoclonal horseradish peroxidase-conjugated anti-IgE antibody (Southern Biotech, Birmingham, AL, USA) at 1:1500 dilution, and chemiluminescence reaction was obtained by ECL (GE Healthcare Life Sciences, Milano, Italy). A ChemiDoc XRS apparatus (Bio-Rad, Milano, Italy) was used for image acquisition.

### 4.7. 2-DE: IPG-SDS-PAGE and 2D IgE-Blotting

WSSF proteins (100 µg) precipitated with 4 volumes of cold acetone were solubilized in 130 µL of rehydration buffer (7 M urea, 2 M thiourea, 20 mM Tris-HCl pH 8, 0.4% CHAPS). Proteins were reduced and alkylated by 1% *w*/*v* DTT and 20 mM iodoacetamide, respectively. ReadyStrips (7 cm, non-linear 3–10 pH range) (Bio-Rad, Milano, Italy) were actively rehydrated for 12 h with the solubilized proteins. Isoelectrofocusing was carried out with Protean IEF Cell (Bio-Rad, Milano, Italy) at 25,000 V/h at 20 °C.

The strips were equilibrated for 30 min in 6 M urea, 30% glycerol, 2% SDS, 50 mM Tris-HCl pH 8.0.

The SDS-PAGE was carried out as above described. After the run the 2 DE gels were either stained with EZBlue Gel Staining Reagent (Sigma, Milano, Italy) or electroblotted.

### 4.8. Electroendosmotic Preparative Gel Electrophoresis (EPGE)

EPGE was performed using an Elettrofor EPGE apparatus (Elettrofor, Rovigo, Italy) [43]. The cylindrical gel (2-cm diameter and 10-cm long) was of 22 mL of 16% polyacrylamide separating gel and 9.5 mL of 4% stacking gel. PBS soluble wheat proteins (5 mg in 2 mL of SDS-PAGE sample buffer containing 5% 2-mercaptoethanol) were loaded onto the gel. The electrophoresis was carried out at 25 mA in 25 mM Tris and 192 mM glycine, finally the eluates were fractionated (2 mL each fraction). The fractions containing proteins of about 21 and 27 kDa were pooled, 20% TCA precipitated and the pellet washed three times with cold acetone to remove SDS. The proteins were then subjected to 2D electrophoresis.

### 4.9. Protein Identification by NanoHPLC Chip MS/MS

The immunodetected proteins were individuated in 2-DE Coomassie stained gels, spots were excised and the in-gel trypsin digestion was performed as described elsewhere [44]. Peptides from 5 μL of each sample were then separated by reversed phase nano-HPLC-Chip technology (Agilent Technologies, Palo Alto, CA, USA) online-coupled with a 3D ion trap mass spectrometer (model Esquire 6000, Bruker Daltonics, Bremen, Germany). Sequential elution of peptides was accomplished using a flow rate of 300 nL/min and a linear gradient from Solution A (2% acetonitrile; 0.1% formic acid) to 50% *v*/*v* of Solution B (98% *v*/*v* acetonitrile; 0.1% *v*/*v* formic acid) in 20 min over the Zorbax 300SB-C18 (40 nL, 5μm) enrichment column and the Zorbax 300SB-C18 (43 mm × 75μm, with a 5-μm particle size) analytical column of Chip. The complete system was fully controlled by ChemStation (Agilent Technologies) and EsquireControl (Bruker Daltonics) software. The scan range used for spectra acquisition was from 300 to 1800 m/z. For tandem MS experiments, the system was operated with automatic switching between MS and tandem MS/MS modes. The three most abundant peptides of each MS scan were selected to be further isolated and fragmented. The MS/MS scanning was performed in the normal resolution mode at a scan rate of 13.000 mass to charge ratio (*m*/*z*) per second. A total of five scans were averaged to obtain an MS/MS spectrum. Database searches were conducted using the MS/MS ion search of Mascot against entries of the SwissProt 2021_03 (565,254 sequences; 203,850,821 residues) with the following parameters: *Viridiplantae* as taxonomy, specific trypsin digestion, up to one missed cleavage; fixed and variable modifications: carbamidomethyl (Cys) and oxidation (Met), respectively; peptide and fragment tolerances: ±0.8 and ±0.6 Da, respectively, and peptide charges: +1, +2, and +3. The protein spots were identified as being a ‘significant hit’ (*p* < 0.05) based on individual peptide ion score.

### 4.10. Wheat Allergen List Update

A review of the scientific literature on wheat flour allergens using free-access on-line databases (Pubmed, Scopus, WOS; key words: “baker’s asthma” OR “baker’s allergy” OR “baker’s rhinitis” AND “wheat allergen”; WHO/IUIS allergen nomenclature database, Allergome, COMPARE, FARRP ALLERGEN PROTEIN DATABASE, AllerBase; key words: OR “wheat allergen” OR “wheat protein” OR “*Triticum aestivum* allergen” OR “*Tricticum aestivum* protein”; last access 31 July 2021) was conducted to update the list of allergens identified to date [25,45,46,47,48].

## 5. Conclusions

The results of the present proteomic study of inhalant wheat allergens suggest the importance to use the wheat flour to which allergic bakers are exposed and the need to identify all the wheat allergens to which bakers are sensitized in order to evaluate their role, by follow-up studies, in bakers suffering from occupational wheat-related allergic comorbidities.

## Figures and Tables

**Figure 1 molecules-27-01212-f001:**
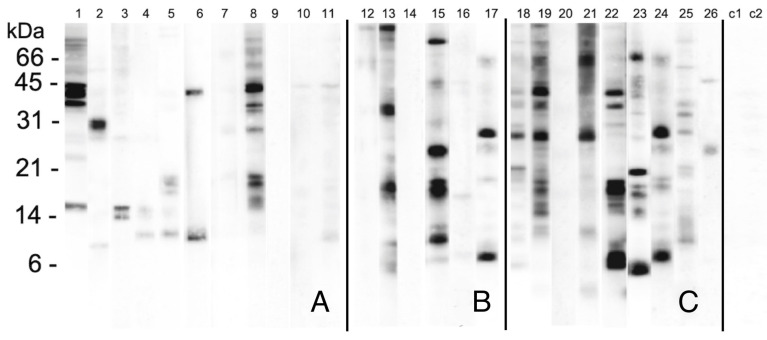
WSSP resolved by SDS-PAGE and probed with the single sera of the bakers with work-related symptoms: rhinitis only (panel (**A**), lanes 1–11), asthma only (panel (**B**), lanes 12–17), asthma and rhinitis (panel (**C**), lanes 18–26) and controls (lane C1 and C2).

**Figure 2 molecules-27-01212-f002:**
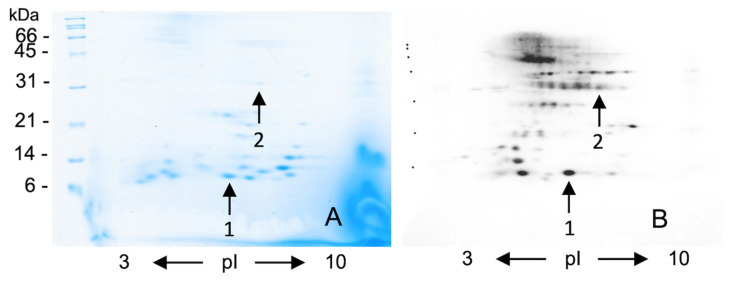
WSSP stained by Coomassie blue 2-DE (panel (**A**)) and immunodetected by IgE-Blotting with the pooled sera of bakers with work-related symptoms (panel (**B**)). The arrows of panel B indicate the spots identified by nanoHPLC-Chip-tandem MS/MS. Spot highlighted by arrow 1 was identified as a monomeric alpha amylase inhibitor, while spot indicated by arrow 2 as a glucose/ribitol dehydrogenase. Further spots with MW in the range of about 21–27 kDa are well visualized by IgE blotting (panel (**B**)) but not stained by Coomassie in 2-DE.

**Figure 3 molecules-27-01212-f003:**
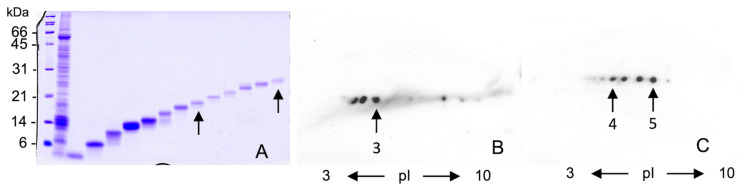
Panel (**A**): 16% SDS-PAGE, Lane 1: MW markers; lane 2: total WSSP, lanes from 3 to 15: fractionated WSSP by electroendosmotic preparative gel electrophoresis. The arrows indicate the fractions resolved by 2-DE and IgE-Blotting (panel (**B**,**C**) respectively). Arrow (3) in panel (**B**) indicates the spot identified as 16.9 kDa class I heat shock protein 1, while the two arrows in panel (**C**) indicate the spots identified as triosephosphate isomerase (4) and thioredoxin peroxidase (5), respectively.

**Table 2 molecules-27-01212-t002:** Demographic and clinical characteristics of work-related wheat allergic bakers and controls.

Patient	Age (Years)	Sex	Smoking	BMI	FEV1 (%)	FEV1/FVC (%)	PD 20 FEV1 (mcg)	Atopy	WHEAT SPT	Wheat IgE (kU/l)	Total IgE (kU/l)	WRS Symptoms
1	54	M	N	32.7	91	91	1074	Y	+	11.0	2408	WRR
2	39	M	N	23.9	95	108	>2000	Y	-	3.6	123	WRR
3	45	M	Y	28.9	118	105	>2000	Y	+	0.6	252	WRR
4	34	M	N	20.6	96	108	>2000	Y	+	5.0	289	WRR
5	42	F	N	19.7	84	93	500	Y	−	2.7	260	WRR
6	34	F	N	21.4	104	98	>2000	Y	−	0.6	49.3	WRR
7	33	M	N	21.1	100	104	>2000	Y	−	22.8	1098	WRR
8	33	M	N	23.9	105	97	>2000	Y	+	0.4	74.4	WRR
9	37	M	Y	26.0	93	93	30	Y	+	3.0	104	WRR
10	39	M	N	29.8	113	99	1939	N	+	15.8	208	WRR
11	32	M	Y	29.0	103	96	>2000	N	+	8.9	138	WRR
12	58	M	N	30.9	118	104	63	Y	+	17.6	885	WRAS
13	53	M	N	38.2	108	104	76	Y	+	38.5	407	WRAS
14	46	M	N	23.2	112	106	>2000	N	+	9.7	44.9	WRAS
15	50	M	N	26.2	122	109	>2000	Y	+	3.6	318	WRAS
16	40	M	N	28.7	100	87	ND	N	−	0.9	63	WRAS
17	35	M	N	23.9	105	97	>2000	Y	+	3.3	106	WRAS
18	55	M	N	22.1	59	96	ND	Y	+	46.9	817	WRAS, WRR
19	29	F	N	24.8	98	93	21	Y	+	47.2	1383	WRAS, WRR
20	33	M	N	49.1	90	84	ND	N	−	0.4	216	WRAS, WRR
21	69	M	N	26.0	97	98	>2000	Y	+	7.3	213	WRAS, WRR
22	37	M	N	30.9	75	87	ND	Y	+	937.0	8291	WRAS, WRR
23	35	M	Y	29.0	103	96	438	Y	+	40.8	1038	WRAS, WRR
24	54	M	N	25.0	110	86	ND	N	+	24.4	1163	WRAS, WRR
25	28	F	Y	33.5	118	115	68	Y	+	28.8	547	WRAS, WRR
26	68	M	N	33.3	83	83	ND	Y	+	407	1594	WRAS, WRR
C1	26	F	N	23.2	112	106	>2000	N	−	<0.35	39.9	−
C2	28	F	Y	22.3	113	123	>2000	N	−	<0.35	<2	−

WRS: Work-Related Symptoms; WRAS: Work-Related Asthma Symptoms; WRR: Work-Related Rhinitis; Y: Yes; N: No; ND: Not Done; + and −: positive negative SPT; C1 and C2: controls.

## Data Availability

The data presented in this study are available on request from the corresponding author.

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
