# Peer review of "Glucose/Ribitol Dehydrogenase and 16.9 kDa Class I Heat Shock Protein 1 as Novel Wheat Allergens in Baker’s Respiratory Allergy"

_molecules, 2022, doi:10.3390/molecules27041212_

Round 1

Reviewer 1 Report

In this paper, Olivieri et al. studied wheat allergens with a mass spectromerty method in Italian bakers. It is exciting that the authors described 2 novel wheat allergens in symptomatic bakers by using their multistep molecular work. There are few published studies about bakers asthma in Italy.

In literature, it was reported that major wheat allergens could be different in different  countries (Ingrid Sander et al. J Allergy Clin Immunol 2015 Jun). In that study recombinant allergens Tria 27,28,29,32 and 39 were the most frequent ones in German, Dutch and Spanish bakers.

On the other hand the newly described proteins are mentioned to have other physiological roles, one is a heat shock protein and other is an enzyme of glucose degradation Therefore, it should be further studied whether these novel allergens are specific to “Italian wheat –Bolero”. Although the number of the study population is quite small to make a comment on this, the authors can write this issue in discussion part.

Also it needs to be clarified that the new allergens are clinically relevant to baker’s asthma , baker’s rhinitis or both.  It is difficult to understand the results of asthma and/or rhinitis from Figure 1. It will be better if the authors can make comments on results of asthma/ rhinitis patients and the possible reasons for differences.

Author Response

Response to Reviewer 1 Comments

In this paper, Olivieri et al. studied wheat allergens with a mass spectromerty method in Italian bakers. It is exciting that the authors described 2 novel wheat allergens in symptomatic bakers by using their multistep molecular work. There are few published studies about bakers asthma in Italy.

Author response: we thank the referee for the appreciation of our work.

In literature, it was reported that major wheat allergens could be different in different  countries (Ingrid Sander et al. J Allergy Clin Immunol 2015 Jun). In that study recombinant allergens Tria 27,28,29,32 and 39 were the most frequent ones in German, Dutch and Spanish bakers.

Author response: we thank the referee for suggesting this further citation that we added at lines 155-156.

On the other hand the newly described proteins are mentioned to have other physiological roles, one is a heat shock protein and other is an enzyme of glucose degradation Therefore, it should be further studied whether these novel allergens are specific to “Italian wheat –Bolero”. Although the number of the study population is quite small to make a comment on this, the authors can write this issue in discussion part.

Author response: we agree with the referee and, in respect with this consideration, we add sentences  (see lanes 171-174 and  200-201).

Also it needs to be clarified that the new allergens are clinically relevant to baker’s asthma , baker’s rhinitis or both.  It is difficult to understand the results of asthma and/or rhinitis from Figure 1. It will be better if the authors can make comments on results of asthma/ rhinitis patients and the possible reasons for differences.

Author response: we partially agree with the Referee and we add a sentence (see lane 171-175) to describe the and explain the different IgE binding patterns between the different groups of patients. On the other hand, as above mentioned, the limited number of subjects involved in this study do not allow further speculations. To highlight this aspect we have add a sentence in lanes 174-175.

Reviewer 2 Report

In this work, Olivieri and collaborators stablished two novel wheat allergens in occupational wheat allergy. They followed several serial steps to identify the allergens. The design of the study is clear and well structured. I only have a few minor comments:

  1. Figure 1: please, highlithed (put a box or similar) each group (rhinitis, asthma and asthma/rhinitis) to differentiate them in a better way.
  2. It is true that inter-individual variability is evident; however, a “similar” profile can be observed in each group of patietns with the same symptoms, for example patients with rhinitis seem to recognize less bands and with lower intensity than individuals with rhinitis and asthma. It could be interesting, and you may discuss and lucubrate this idea in discussion section.
  3. It could be very interesting that authors perform a western blot with new allergens in a separate manner (purified allergens) and in each individual patient. This fact could elucidate how many patients (or which of them) could recognize the new allergen/s. Is it possible to do this?

Author Response

Response to Reviewer 2 Comments

In this work, Olivieri and collaborators stablished two novel wheat allergens in occupational wheat allergy. They followed several serial steps to identify the allergens. The design of the study is clear and well structured.

Author response: we thank the referee for the appreciation of our work.

I only have a few minor comments:

    Figure 1: please, highlithed (put a box or similar) each group (rhinitis, asthma and asthma/rhinitis) to differentiate them in a better way.

Author response: done. We hope that in the revised form the presentation of figure 1 resulted improved.

    It is true that inter-individual variability is evident; however, a “similar” profile can be observed in each group of patietns with the same symptoms, for example patients with rhinitis seem to recognize less bands and with lower intensity than individuals with rhinitis and asthma. It could be interesting, and you may discuss and lucubrate this idea in discussion section.

Author response: we agree with the Referee and, under this point of view we have added a sentence (please see lanes 173-176). Unfortunately the scarce number of subjects participating at this research do not allow further speculations (see lanes 176-177).

    It could be very interesting that authors perform a western blot with new allergens in a separate manner (purified allergens) and in each individual patient. This fact could elucidate how many patients (or which of them) could recognize the new allergen/s. Is it possible to do this?

Author response: we correspond with the referees that the use of single patient sera could elucidate the role of each allergen in a specific sensitization. Unfortunately this approach requires, in particular for 2D analyses, the availability of large amount of sera and this is scarcely compatible with an ethical approach and our technical possibilities. To describe this, a sentence was added (see lanes 211-214).
